# Towards single-chip radiofrequency signal processing via acoustoelectric electron–phonon interactions

Lisa Hackett[1], Michael Miller[1], Felicia Brimigion[1], Daniel Dominguez[1], Greg Peake[1], Anna Tauke-Pedretti[1], Shawn Arterburn[1], Thomas A. Friedmann[1] & Matt Eichenfield [1 ✉]

The addition of active, nonlinear, and nonreciprocal functionalities to passive piezoelectric acoustic wave technologies could enable all-acoustic and therefore ultra-compact radio-frequency signal processors. Toward this goal, we present a heterogeneously integrated acoustoelectric material platform consisting of a 50 nm indium gallium arsenide epitaxial semiconductor film in direct contact with a 41° YX lithium niobate piezoelectric substrate. We then demonstrate three of the main components of an all-acoustic radiofrequency signal processor: passive delay line filters, amplifiers, and circulators. Heterogeneous integration allows for simultaneous, independent optimization of the piezoelectric-acoustic and electronic properties, leading to the highest performing surface acoustic wave amplifiers ever developed in terms of gain per unit length and DC power dissipation, as well as the first-ever demonstrated acoustoelectric circulator with an isolation of 46 dB with a pulsed DC bias. Finally, we describe how the remaining components of an all-acoustic radiofrequency signal processor are an extension of this work.

[1] Microsystems Engineering, Science, and Applications, Sandia National Laboratories, Albuquerque, NM, USA. ✉email: meichen@sandia.gov

Modern radiofrequency (RF) signal processors (RFSPs) make use of multiple technologies, integrated at the system level, to achieve all their necessary functionalities. For example, high-performance passive filtering and delay lines are typically achieved in piezoelectric acoustic wave technologies such as lithium niobate (LiNbO$_3$) surface acoustic wave (SAW)[1–3] and aluminum nitride bulk acoustic wave (BAW)[4,5] platforms. Active devices such as amplifiers and mixers are typically realized in semiconductor technologies such as a complementary metal oxide semiconductor (CMOS)[6,7] and gallium arsenide (GaAs)[8,9]. Nonreciprocity for circulators and isolators is achieved through the gyromagnetic effect via magnetized ferrite material in proximity to RF transmission lines[10,11]. More recent approaches to nonreciprocal RF devices use spatiotemporal modulation in coupled inductor-capacitor (LC) or acoustic resonators in a ring or wye topology, but require an RF drive for operation[12–15].

While these technologies are all highly optimized for the performance of their respective components, their integration at the system level ultimately hinders the miniaturization of RFSPs, which is in direct conflict with the overarching technological trend toward simultaneous proliferation and miniaturization of RF systems[16]. The complexity of the RFSP front-end—especially in terms of the number of required filters and amplifiers—will increase significantly with the requirement to support more frequency bands[17,18] and multiple-input/multiple-output designs[19] in an increasingly crowded and congested spectral landscape. In addition, simultaneous transmit and receive (STAR) systems where the transmitter (Tx) and receiver (Rx) operate simultaneously in the same frequency band are highly sought after to improve spectral efficiency by essentially doubling the available bandwidth[20]. An all-acoustic RFSP would be ultra-compact, but fundamental components remain to be demonstrated. SAW signal processing can also address a wide range of frequencies from MHz to GHz by simply changing the electrode pitch[21–23]. This could allow the RFSP of *many* frequency bands to be fabricated on a single chip, greatly reducing the size, the number of

components, packaging requirements, and assembly required to achieve the complex RFSP circuitry present in modern RF devices.

In this article, we demonstrate three of the necessary fundamental components, fabricated on the same chip, to realize an all-acoustic and therefore ultra-compact RFSP: passive filtering with time delay, amplification, and non-reciprocal circulators, and, in the discussion, we lay out a vision for an all-acoustic RFSP based on the acoustoelectric effect. We present the best performing acoustic amplifier ever in terms of gain per unit length and DC power dissipation, the first-ever acoustoelectric circulator, and the co-fabrication of these active acoustic wave components on the same chip as high performing passive acoustic wave delay lines. We achieve this via heterogeneous integration of piezoelectric-acoustic wave and semiconductor materials to produce strong acoustoelectric electron-phonon interactions between the electronic charge carriers in the semiconductor and the electric fields associated with the SAW phonons. Using this platform, we simultaneously realize the exquisite performance of passive piezoelectric acoustic wave devices and selectively endow acoustic circuit elements with nonlinear and nonreciprocal functionality through electron-phonon acoustoelectric interactions[24–29]. Our heterogeneously integrated acoustoelectric platform consists of a 50 nm epitaxial indium gallium arsenide (In$_{0.53}$Ga$_{0.47}$As) semiconductor layer on a 41° YX LiNbO$_3$ substrate that supports a shear-horizontal SAW (SH-SAW) with an electromechanical coupling coefficient ($K^2$) of 17% and minimal propagation losses[30]. We achieve an acoustically thin, high mobility In$_{0.53}$Ga$_{0.47}$As semiconductor layer (measured Hall mobility of 2000 cm$^2$/V-s); with a thickness of only 50 nm, the devices can provide acoustoelectric interactions out to at least 5 GHz[31]. Furthermore, the low silicon (Si) doping density of electrons, $N$, in the semiconductor ($1 \times 10^{16}$ cm$^{-3}$), made possible due to metal-organic chemical vapor deposition (MOCVD), provides extremely high gain with low ohmic heat dissipation. In addition to passive delay lines on 41° YX LiNbO$_3$, we demonstrate a 276 MHz acoustic wave amplifier with a terminal gain of 13.5 dB in

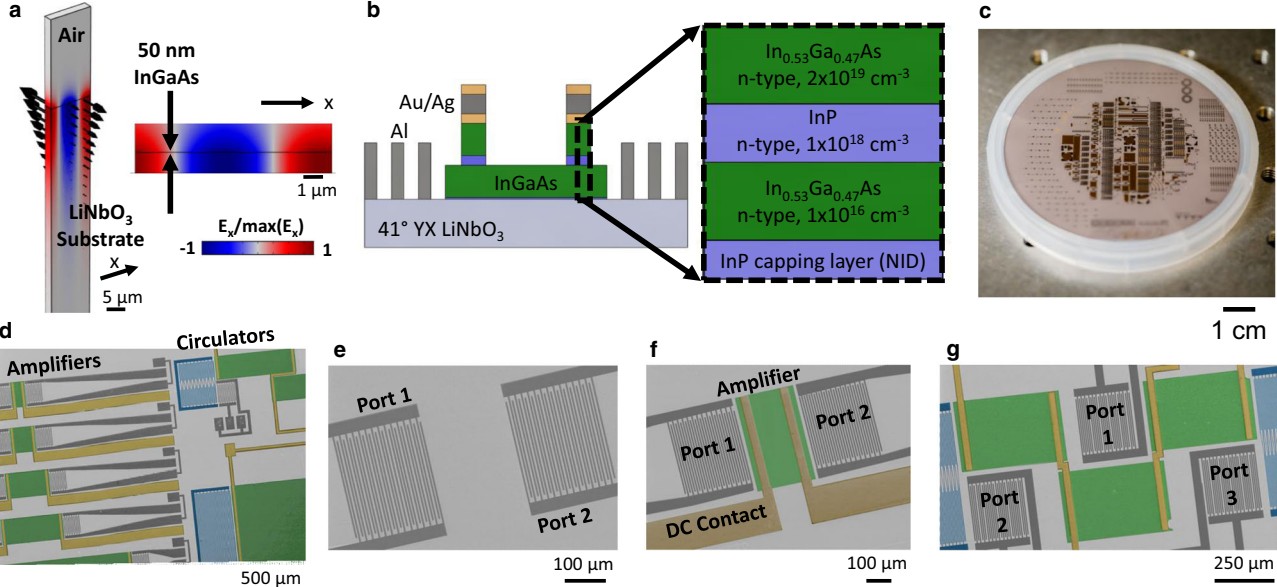

**Fig. 1 Material platform overview. a** Longitudinal electric field ($E_x$) from a finite element model, with displacement field indicated by the black arrows, and (**b**) a schematic of our implementation of the acoustoelectric effect for an In$_{0.53}$Ga$_{0.47}$As thin film on a LiNbO$_3$ substrate. **c** Image of the heterostructure wafer after processing. **d** False-colored SEM images of a wafer with passive and active acoustic wave fabricated devices including (**e**) delay lines, (**f**) amplifiers, and (**g**) circulators. In the false-colored images, the LiNbO$_3$ substrate is light gray, the In$_{0.53}$Ga$_{0.47}$As thin film is green, the RF transducers are dark gray, the DC contacts are gold, and the RF couplers are blue.

505 μm that dissipates only 15 mW of DC power and the first-ever nonmagnetic acoustoelectric circulator that requires only a DC bias for operation with an isolation of 46 dB.

## Results

**Material platform overview.** By heterogeneously integrating and selectively removing epitaxial semiconductor layers on a piezo-electric substrate, we selectively create active and nonreciprocal acoustic devices through the acoustoelectric effect on the same wafer on which we fabricate high performance passive piezo-electric acoustic devices. The acoustoelectric effect arises in our platform due to charge bunching mediated by the interaction between charge carriers in the semiconductor and the evanescent electric field that accompanies the piezoelectric SAW[26]. Depending on the charge carrier drift velocity $v_d$, controlled by an applied drift field $E_d$, the bunched charge either lags or leads the acoustic wave. The gain mechanism, which is analogous to an electro-magnetic traveling-wave tube amplifier[32], depends on the average interaction of the charge carriers with the piezoelectric electric field. This interaction controls the energy transfer from or to the acoustic wave, resulting in acoustic wave attenuation or amplification, respectively. The simulated longitudinal electric field and mechanical displacement for the system we study here, which is a 50 nm $In_{0.53}Ga_{0.47}As$ thin film on a 41°YX $LiNbO_3$ substrate supporting an SH-SAW, is shown in Fig. 1a. The evanescent electric field of the piezoelectric acoustic wave has a strong overlap with the $In_{0.53}Ga_{0.47}As$ thin film in this material system resulting in strong coupling between the electrons and phonons.

While detailed models of acoustic gain and nonreciprocity can give a complete picture of the electron–phonon interaction[33,34], under the appropriate approximations these models are well-captured by a simple, intuitive model that can be derived from an equivalent RC circuit[35]. The acoustic loss $\alpha$ is given by

$$\alpha = \frac{1}{2} k_{AE}^2 k_0 \frac{\gamma \omega \tau}{1 + (\gamma \omega \tau)^2} \qquad (1)$$

where $k_{AE}^2$ is a coupling coefficient describing the interaction strength between the free electrons and the electric field of the piezoelectric acoustic wave, $k_0$ is the acoustic wavenumber, $\gamma = 1 - (v_d/v_a)$, $v_a$ is the acoustic velocity, $v_d = \mu E_d$ where $\mu$ is the mobility, and $\tau$ is given by $\tau = \frac{\varepsilon_0 + \varepsilon_p}{k_0 \sigma t}$ where $\sigma t$ is the semiconductor conductivity-thickness product, $\varepsilon_0$ is the permittivity of free space, and $\varepsilon_p$ is the piezoelectric permittivity. The gain of a material system is then parameterized by only $k_{AE}^2$, $\mu$, and $\sigma t$ where $k_{AE}^2$ is equivalent to $K^2$. Important properties to note are that negative $\alpha$, or acoustic gain, only occurs when $v_d$ exceeds $v_a$ meaning that a voltage $V_{0\,dB}$ must be applied to reach 0 dB. The acoustoelectric interaction is then inherently non-reciprocal as the acoustic wave must travel in the same direction as the unidirectional charge carrier drift to be amplified. In addition, the maximum achievable loss or gain for a given operating frequency is $\alpha_{max} = k_{AE}^2/4$ and therefore only depends on $k_{AE}^2$.

A schematic of our implementation of the acoustoelectric effect as an acoustic amplifier is shown in Fig. 1b. The material platform consists of a heterostructure of an epitaxial 50 nm $In_{0.53}Ga_{0.47}As$ semiconductor, grown by MOCVD, bonded and processed on the wafer-scale to a 41° YX $LiNbO_3$ substrate. The $In_{0.53}Ga_{0.47}As$ can be made acoustically thin when combined with our process where thicker layers with various doping concentrations, $N$, including layers that are not intentionally doped (NID), are selectively removed. Figure 1c shows an image of the $In_{0.53}Ga_{0.47}As/LiNbO_3$ heterostructure wafer following fabrication while Fig. 1d shows a false-colored SEM image of acoustic amplifiers and circulators.

Excluding the $In_{0.53}Ga_{0.47}As$, by selectively etching it away, enables delay line filters and resonators in the underlying piezoelectric technology. Additional false-colored SEM images of an acoustic delay line, amplifier, and circulator are shown in Fig. 1e–g, respectively (see Supplementary Note 1 and Supplementary Figs. 1–2 for additional details on the fabrication process flow and optical microscope device images).

While high-performance delay line filters are already ubiquitous in RF signal processing, here we demonstrate, on the same chip, high-performance acoustoelectric amplifiers and the first-ever acoustoelectric circulator. The performance demonstrated herein—in terms of terminal gain, nonreciprocity, and dissipated DC power—is achieved due to the combination of exceptional piezoelectric and semiconductor material parameters, achievable here for the first time in a heterogeneously integrated material platform.

The rest of the article proceeds as follows. We begin by discussing the optimization of the electromechanical and acoustoelectric interactions. We then present the three devices experimentally demonstrated here: an acoustic delay line, acoustoelectric amplifier, and acoustoelectric circulator. We then conclude with a discussion of the remaining components necessary to achieve an all-acoustic RFSP and other potential applications of the technology, such as integrated photonics.

**Optimization of the electromechanical and acoustoelectric interaction.** To operate as a platform for passive and active acoustic wave devices, the material properties of the $In_{0.53}Ga_{0.47}As$ on $LiNbO_3$ heterostructure must be optimized for acoustic wave generation, amplification, and nonreciprocity. All components are improved by a piezoelectric substrate with a large $K^2$ for surface-guided waves and low propagation loss. This leads to lower insertion losses and larger operating bandwidths in addition to a larger $k_{AE}^2$. For the acoustic modes of interest on commercially available $LiNbO_3$ substrates, the SH-SAW mode on 41° YX $LiNbO_3$ has a $K^2$ of 17%, is lossless for the open boundary condition, and has a propagation loss of only 0.04 dB/Λ for the shorted boundary condition, where Λ is the acoustic wavelength (see Supplementary Note 2 for a summary of the properties of different $LiNbO_3$ substrates). Due to the exceptional $K^2$ and relatively low propagation losses, we focus our work on this SH-SAW mode (see Supplementary Note 3 and Supplementary Fig. 3 for a comparison of $k_{AE}^2$ between the Rayleigh mode on YZ $LiNbO_3$ and the SH-SAW on 41°YX $LiNbO_3$).

The importance of the piezoelectric and semiconductor material parameters for acoustoelectric device performance is quantified in Fig. 2a–c. Overall, to improve the gain and DC power dissipation, it is most important to maximize $K^2$ and minimize $\sigma t$. Best performance is obtained by maximizing the gain slope, which, from Eq. (1), requires optimization of $K^2$, $\sigma t$, and $\mu$. However, minimizing the DC power dissipation requires a different optimization with respect to $\sigma t$ and $\mu$. Figure 2a and b show contour plots of the gain slope in dB/V as a function of {$\sigma t$ and $K^2$} and {$\sigma t$ and $\mu$}, respectively. The highest gain slope is achieved when $K^2$ is maximized and $\sigma t$ is minimized while increasing $\mu$ has no impact only if there is no corresponding increase in $\sigma t$. To maintain the same gain slope with a larger $\mu$, $\sigma t$ must be reduced by a lower $N$ or thinner semiconductor film. Figure 2c shows a contour plot of the dissipated DC power ($P_d$) to achieve 25 dB of gain as a function of $\sigma t$ and $\mu$ for an acoustoelectric amplifier with a length ($l$) of 500 μm and width ($w$) of 328 μm where $P_d = \frac{V^2 w \sigma t}{l}$. For low $\sigma t$, $P_d$ can be further improved by increasing $\mu$, which decreases $V_{0dB}$. As can be seen from Fig. 2c, there is a range of $\mu$ values that minimize $P_d$ for a given $\sigma t$. In our material platform, we achieve a $K^2$ of 17%, a $\sigma t$ of

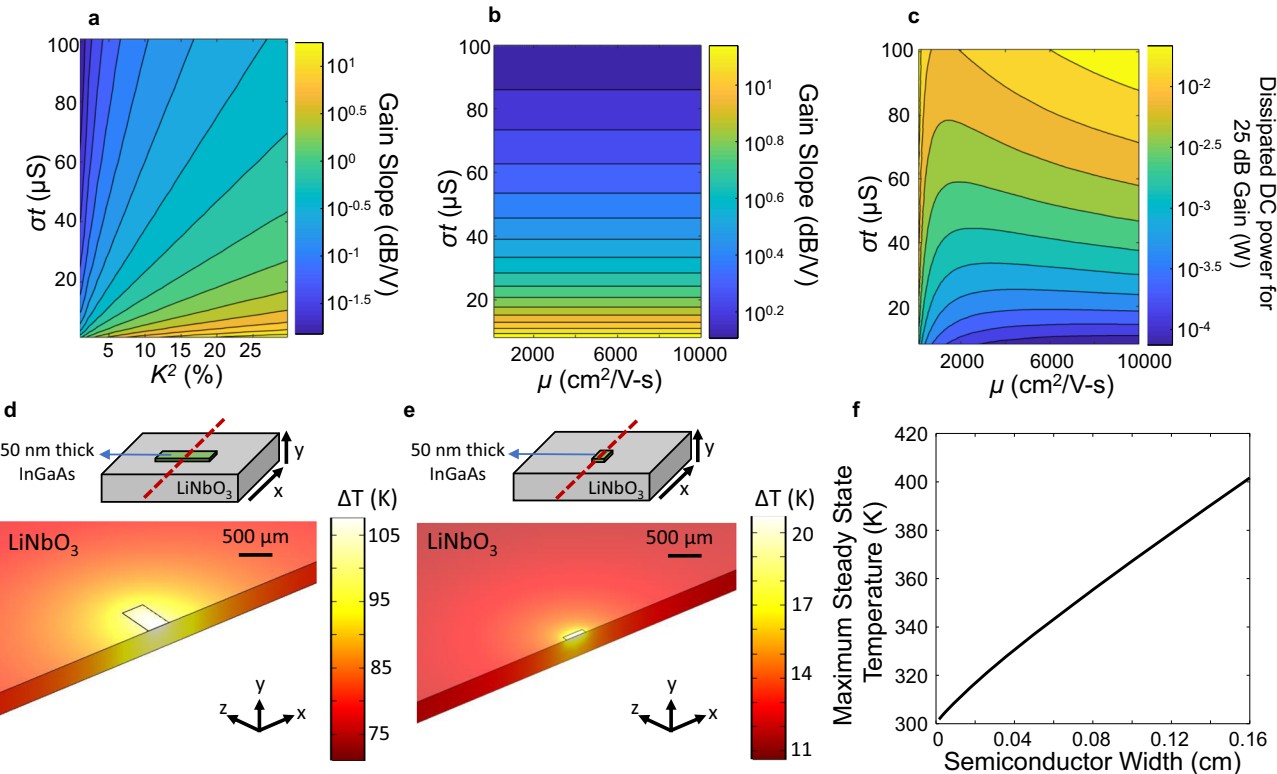

**Fig. 2 Material parameters and geometry optimization.** Contour plots of the (**a**) gain slope as a function of σt and $K^2$ (**b**) gain slope as a function of σt and μ, and (**c**) dissipated DC power for 25 dB of gain as a function of σt and μ. The simulated steady-state distribution of the temperature difference (ΔT) is shown for a semiconductor width of (**d**) 0.16 cm and (**e**) 0.024 cm where the dashed-red line denotes the axis for both the propagation of the acoustic wave and the simulated cross-section along with (**f**) a plot of the maximum steady-state temperature as a function of semiconductor width from 0.0016 cm to 0.16 cm.

16 µS, and mobility of 2000 cm²/V-s. The specially selected SH-SAW mode on 41°YX LiNbO₃ with high $K^2$ strongly couples the acoustic wave to semiconductor charge carriers while simultaneously having low radiation propagation loss from coupling to bulk modes. Heterogeneous integration of the materials allows for an extremely large interaction strength to be combined with highly optimized semiconductor performance.

Another important consideration for active acoustic wave devices is thermal management, which has historically limited the duty cycle with which amplifiers can be operated[24,36]. The semiconductor layer gives rise to Joule heating and the heat dissipation efficiency depends on convection into the air and thermal conduction into the underlying substrate. The simulated steady-state temperature distribution is shown for a 50 nm thick In₀.₅₃Ga₀.₄₇As semiconductor layer with a length of 0.05 cm and widths of 0.16 cm (Fig. 2d) and 0.024 cm (Fig. 2e) on a LiNbO₃ chip with a length, width, and thickness of 12.5 mm × 24 mm × 0.5 mm, respectively, substrate thermal conductivity of 4.2 W/(mK), and a heat transfer coefficient of 5 W/(m²K). The simulated maximum steady-state temperature as a function of semiconductor width is shown in Fig. 2f. A less wide semiconductor layer is both more resistive–and therefore a smaller overall heat source–and provides more efficient lateral heat dissipation as the heat is efficiently conducted away in the transverse direction. There are situations where the semiconductor width cannot be significantly reduced, for example, because a large aperture is required to achieve a large, diffraction-free propagation length in a delay line. In that case, heat dissipation could also be significantly improved by increasing the thermal conductivity of the substrate, such as using LiNbO₃ thin film on Si instead of a bulk LiNbO₃ piezoelectric.

**Acoustic delay line.** High-performance delay lines were fabricated on LiNbO₃ using single-phase unidirectional transducers (SPUDT). Acoustic delay lines with low insertion loss and large operating bandwidth are of high interest for RF applications due to their compact size and ability to incorporate novel passive signal processing functions through the design of the transducer transfer function. Figure 3a shows a schematic of a SAW delay line that uses two traditional interdigitated transducers (IDTs), which have a pitch of Λ/2 and an electrode width of Λ/4. Figure 3b shows a schematic of a SAW delay line that uses two SPUDTs. The traditional IDT results in bidirectional acoustic wave generation, as shown in the simulated displacement field in Fig. 3c, which leads to a minimum insertion loss of 6 dB in a delay line. The SPUDT design uses integrated reflectors with spacings from Λ/8 transduction electrodes such that there is constructive interference for the forward propagating wave and destructive interference for the backward propagating wave. As a result, the generation and detection of acoustic waves are unidirectional, ideally reducing the insertion loss by 3 dB per transducer, as shown in the simulated displacement field in Fig. 3d.

The measured S₂₁ as a function of frequency for a delay line on YZ LiNbO₃ supporting a Rayleigh SAW mode with SPUDT transducers is shown in Fig. 3e. The delay line is designed with Λ = 16 µm, an aperture of 25Λ, propagation length of 500 µm, and 45 SPUDT cells. The experimental resonant frequency is 214 MHz, the insertion loss is 4.4 dB, and the fractional bandwidth (FBW) is 1.5%. A plot of the measured S₂₁ as a function of frequency for a delay line on 41° YX LiNbO₃ supporting an SH-SAW mode with SPUDT transducers is shown in Fig. 3f. The delay line has Λ = 16 µm, an aperture of 15Λ, propagation length of 250 µm, and 21 SPUDT cells. The resonant frequency is 276

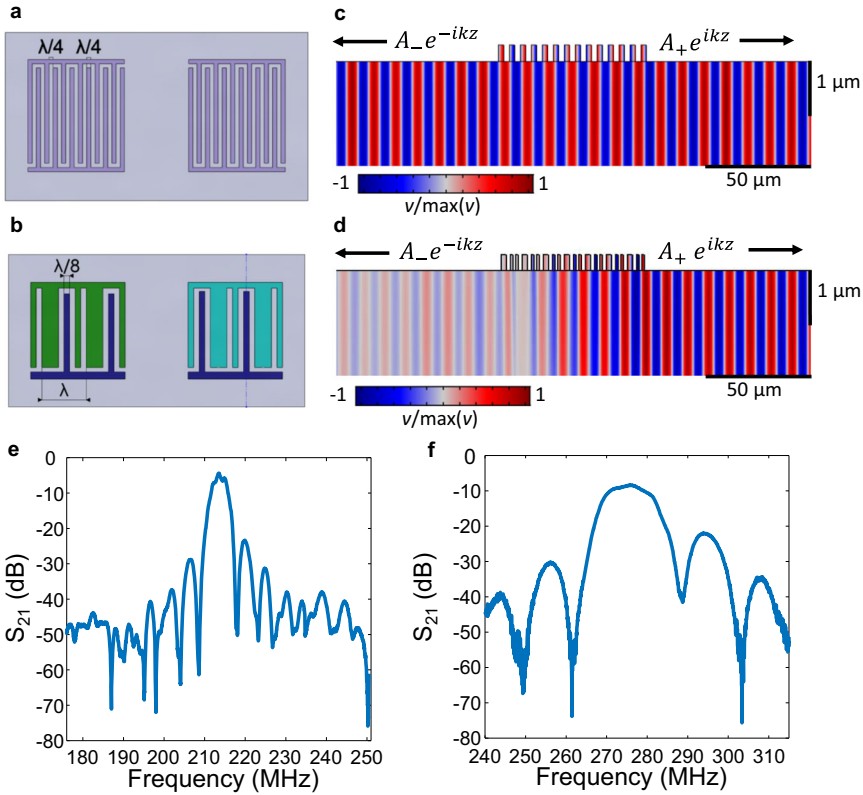

**Fig. 3 Acoustic delay line.** Schematics of a (**a**) traditional IDT delay line and a (**b**) SPUDT design with corresponding simulated normalized displacement fields shown in **c** and **d**, respectively. The LiNbO$_3$ substrate is shown in light gray, the aluminum for the traditional IDTs is shown in light purple, and the aluminum for the SPUDT IDTs is shown in dark purple, green, and cyan. $\lambda$ denotes the acoustic wavelength, $A_- e^{-ikz}$ describes the backward propagating wave, $A_+ e^{ikz}$ describes the forward propagating wave, and $v$ is the displacement along the $y$-axis. Measured S$_{21}$ as a function of frequency for SPUDT delay lines on (**e**) YZ LiNbO$_3$ supporting a Rayleigh SAW mode with a resonant frequency of 214 MHz and (**f**) 41° YX LiNbO$_3$ supporting an SH-SAW mode with a resonant frequency of 276 MHz.

MHz, the insertion loss is 8.4 dB, and the FBW is 4.1%. For the delay lines on YZ and 41° YX LiNbO$_3$, matching to the 50 Ω source impedance of the network analyzer was optimized through a two-step process. The approximate number of electrode pairs was calculated using the Mason equivalent circuit model given the IDT aperture, expected capacitance, $K^2$, and operating frequency[37]. A bank of delay lines was then fabricated with a varying number of electrode pairs followed by measuring the S-parameters as a function of the number of electrode pairs to determine the optimal number. Previous results have indicated that a more sophisticated transducer design is required to achieve low insertion loss in SH-SAW delay lines compared to Rayleigh waves due to effects such as bulk wave interference[38]. Overall performance for a SPUDT transducer delay line is limited by the fundamental tradeoff between insertion loss and FBW. Ideally, $K^2$ and the reflectivity per unit cell will be maximized such that impedance matching and unidirectionality can be achieved with a smaller number of electrodes, which will maximize the FBW while maintaining a low insertion loss. An optimized transducer design will ultimately result in our devices having lower insertion loss that leads to higher performance.

**Acoustic amplifier.** Voltage-controlled amplification and attenuation of the SH-SAW mode on 41° YX LiNbO$_3$ are accomplished by adding a patterned In$_{0.53}$Ga$_{0.47}$As epitaxial semiconductor with DC contacts to the RF acoustic delay line. The experimental setup for gain measurements is shown in Fig. 4a. The gain is measured by detecting the change in S$_{21}$ with respect to time during a continuous RF input at 276 MHz and the application of a 1 ms voltage pulse to the amplifier. The S$_{21}$ value represents the power transmission from the input to the output. Pulsed mode operation is used to avoid thermal drift and runaway, which can corrupt gain measurements. For the acousto-electric amplifier, the gain bandwidth is limited by the transducer bandwidth.

Here we present our experimental gain data in two ways. One is the electronic gain, which can only be achieved if the amplification is large enough to overcome losses due to the acoustoelectric effect. The other type of gain we present is terminal gain, which can only be achieved once the amplification is large enough to overcome all of the losses in the system, including transducer conversion losses and reflection losses from the thick DC contact metal. Both electronic gain and terminal gain are gains in power (as opposed to amplitude) measured on a network analyzer with a source impedance of 50 Ω. A plot of electronic gain in dB per unit length as a function of drift field is shown in Fig. 4b for two devices with a 50 nm In$_{0.53}$Ga$_{0.47}$As amplifier layer and acoustoelectric interaction lengths of 255 μm and 505 μm.

The theoretical gain curve, calculated from Eq. (1), is also shown in Fig. 4(b). The theoretical gain curve is calculated based on a $K^2$ of 17%, $v_a$ of 4416 m/s, operating frequency of 276 MHz, $\mu$ of 2000 cm$^2$/V-s, and an $N$ of $1 \times 10^{16}$/cm$^3$. The theoretical prediction fits the experiment well at drift fields less than −0.74 kV/cm, which confirms the individually measured LiNbO$_3$ and In$_{0.53}$Ga$_{0.47}$As material parameters. However, the model fails at larger drift fields, indicating that the experimental gain rollover cannot be captured with a model, such as that from Eq. (1), that is valid only for small signal, steady-

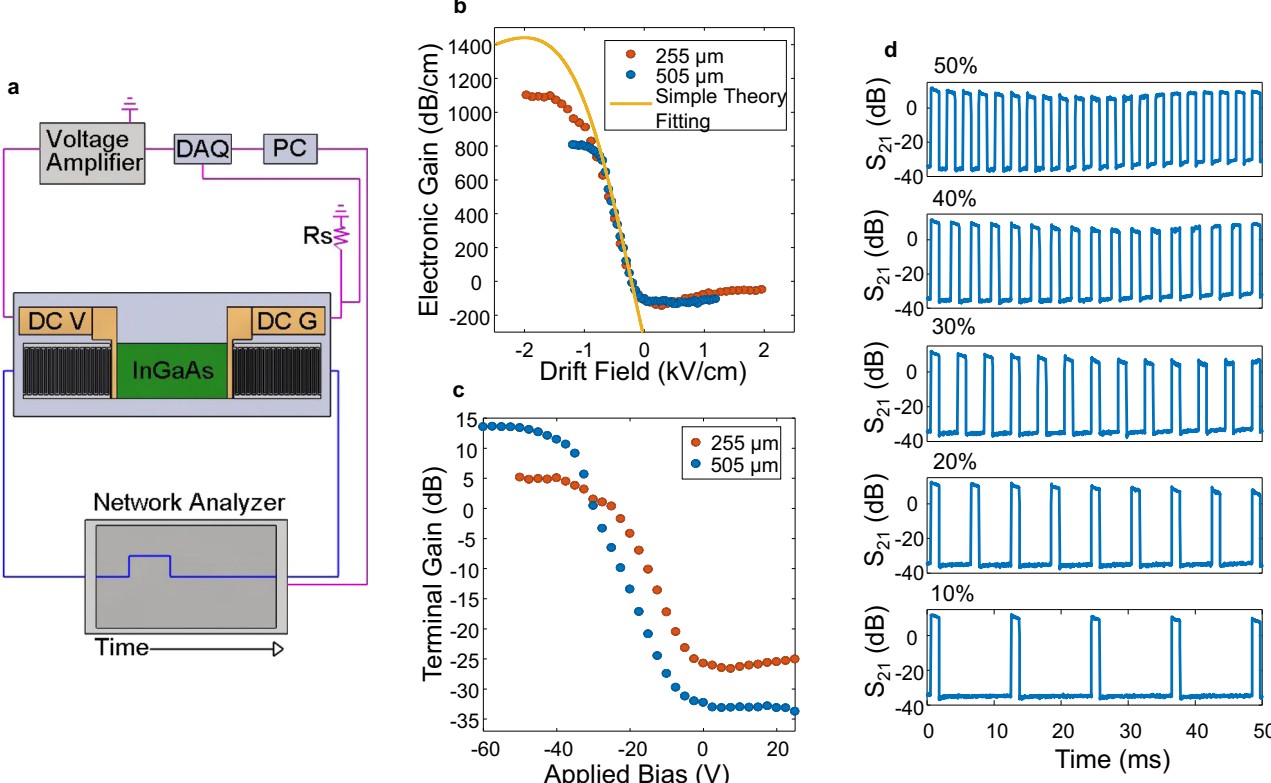

**Fig. 4 Acoustoelectric amplifier. a** Schematic of the experimental setup for gain measurements consisting of a computer (PC), network analyzer, voltage amplifier, data acquisition system (DAQ), and shunt resistor (Rs). DC V and DC G refer to the applied DC bias and ground, respectively. **b** Measured experimental gain as a function of drift field with a comparison to the expected electronic gain from Eq. (1). **c** Terminal gain as a function of applied bias. A terminal gain of 13.5 dB is achieved at −60 V applied bias for a 505 μm long amplifier device. **d** Experimental gain pulses with increasing applied bias duty cycles from 10% to 50%.

state conditions where the material parameters are spatially uniform and do not change with voltage or temperature. The 255 μm long device has a maximum electronic gain of 1101 dB/cm at a drift field of −2 kV/cm and the 505 μm long device has a maximum electronic gain of 806 dB/cm at a drift field of −1.2 kV/cm. Although we demonstrate the largest electronic gains per unit length for any known acoustoelectric material platform, the performance of these devices is limited by thermal effects and can be further improved to operate continuously and reach the true peak acoustoelectric gain by engineering the system to dissipate heat more effectively.

The terminal gain of the acoustoelectric amplifier with increasing duty cycles was examined in order to further assess the acoustoelectric device performance. A plot of terminal gain in dB as a function of applied bias for a 1 ms voltage pulse is shown in Fig. 4c, for the 505 μm long acoustoelectric amplifier. A terminal gain of 13.5 dB is obtained at an applied bias of −60 V, corresponding to a dissipated DC power of only 15 mW. The thermal stability of the acoustoelectric amplifier was investigated by increasing the duty cycle of the 1 ms voltage pulse from 10% to 50% during a 50 ms measurement cycle. Gain measurements with respect to time, taken during the application of the voltage pulses, are shown in Fig. 4d for the increasing duty cycles. The gain remains stable at a 50% duty cycle, with over 10 dB of terminal gain demonstrated.

We also measured the acoustic output power and DC to RF acoustic power efficiency as a function of the DC power dissipation and acoustic input power (see Supplementary Note 4 and Supplementary Fig. 4). The power efficiency ($\eta$) is defined as $\eta = \frac{P^A_{OUT}}{P^{DC}_{DISS}}$ where $P^A_{OUT}$ is the acoustic output power and $P^{DC}_{DISS}$ is the

dissipated DC power. We achieve an $\eta$ of 4.4% at an acoustic input power of −34 dBm and a DC power dissipation of 12 mW. Ultimately $\eta$ is limited by the fact that signal saturation of an acoustic amplifier fundamentally occurs when all the electrons available for the acoustoelectric interaction are incorporated into the acoustoelectrically induced RF current, saturating the gain[39]. The saturation output power increases with N, but this will also increase $P^{DC}_{DISS}$. Depending on device requirements, $\eta$ can potentially be optimized by exploring the parameter space of N, $\sigma t$, and RF saturation power with device designs that prioritize efficient thermal dissipation. The noise figure of acoustoelectric amplifiers has been discussed in previous works and it was found that trapping effects in the semiconductor are the dominant source of noise[40]. Devices using evaporated semiconductor materials had a defectivity-limited noise figure of 9 dB[41]. In the limit of direct contact between the semiconductor and piezoelectric and very small trap density, as expected for an MOCVD film, the theoretical noise figure for the experimental values in this work is 1 dB, under the assumption that trapping effects are the dominant noise source (see Supplementary Note 5).

The critical obstacles to integrate acoustoelectric devices into future RFSPs are stable operations with a continuously applied DC bias, in addition to increasing the RF input power dynamic range and overall RF power handling. Recent experiments show that, by using a LiNbO$_3$ thin film on a Si substrate (as opposed to a bulk LiNbO$_3$ substrate) that still supports guided acoustic waves with and without the amplifier material, the heat dissipation efficiency drastically improves via the 30X improvement in the thermal conductivity of Si compared to LiNbO$_3$[42,43]. The device

fabrication remains identical except for replacing the LiNbO$_3$ substrate with a LiNbO$_3$ on Si substrate. In a 250 μm long device, we achieve an electronic gain of 15.4 dB, while operating continuously and output power of 8.7 dBm (see Supplementary Note 6 and Supplementary Fig. 5). These initial results suggest that the LiNbO$_3$ film on bulk Si substrate is a promising path forward to overcome the challenges of thermal management and power handling for active acoustic wave devices, and the material and device parameters will be further optimized in our future work.

**Acoustoelectric circulator.** As discussed above, the acoustoelectric effect is inherently nonreciprocal. Thus, a circulator can in principle be formed by incorporating the effect into a three-port device with proper management of the direction of gain between each port. A schematic of the developed three-port acoustoelectric circulator is shown in Fig. 5a. Reversing multistrip couplers (RMSCs)[44,45] serve to transmit surface acoustic waves in a designed frequency band from an upper track to a lower track, while simultaneously reversing the propagation direction. These RMSCs consist of periodic metal electrodes that are interlaced between two spatially distinct but parallel channels. Due to the connections between electrodes in the upper and lower channels, the complex electric potentials that are generated in the electrodes of one channel of the RMSC by the SAW generate potentials in

the other channel that excite a SAW propagating in the opposite direction. RMSCs have been previously demonstrated for acoustic ring filters, as shown in Fig. 5b, where they function to transmit the SAW energy generated at an IDT in an upper track to be detected at an IDT in a lower track with minimal acoustic losses[46].

For the acoustoelectric circulator demonstrated here, at each of the three IDTs, which form the three ports of the circulator, an electrical signal can be converted to an acoustic wave by the inverse piezoelectric effect. This acoustic wave is then amplified only for one helicity depending on the sign and magnitude of the applied drift field. Simultaneously, the acoustic wave for the opposite helicity is attenuated. Here helicity defines the two ways that the SAW can travel around the loop of the three ports connected by the RMSCs. One helicity then defines counter-clockwise propagation of the SAW while the other helicity defines clockwise SAW propagation. Once the acoustic wave reaches the sequential port's IDT, it is converted back into an electrical signal by the piezoelectric effect. The theoretical nonreciprocity as a function of applied bias for each acoustoelectric section of the device is shown in Fig. 5c. As can be seen, the contrast ratio between forward and backward propagating waves increases with applied bias. We then seek to achieve an acoustoelectric circulator with low insertion loss and large isolation while maintaining low dissipated DC power.

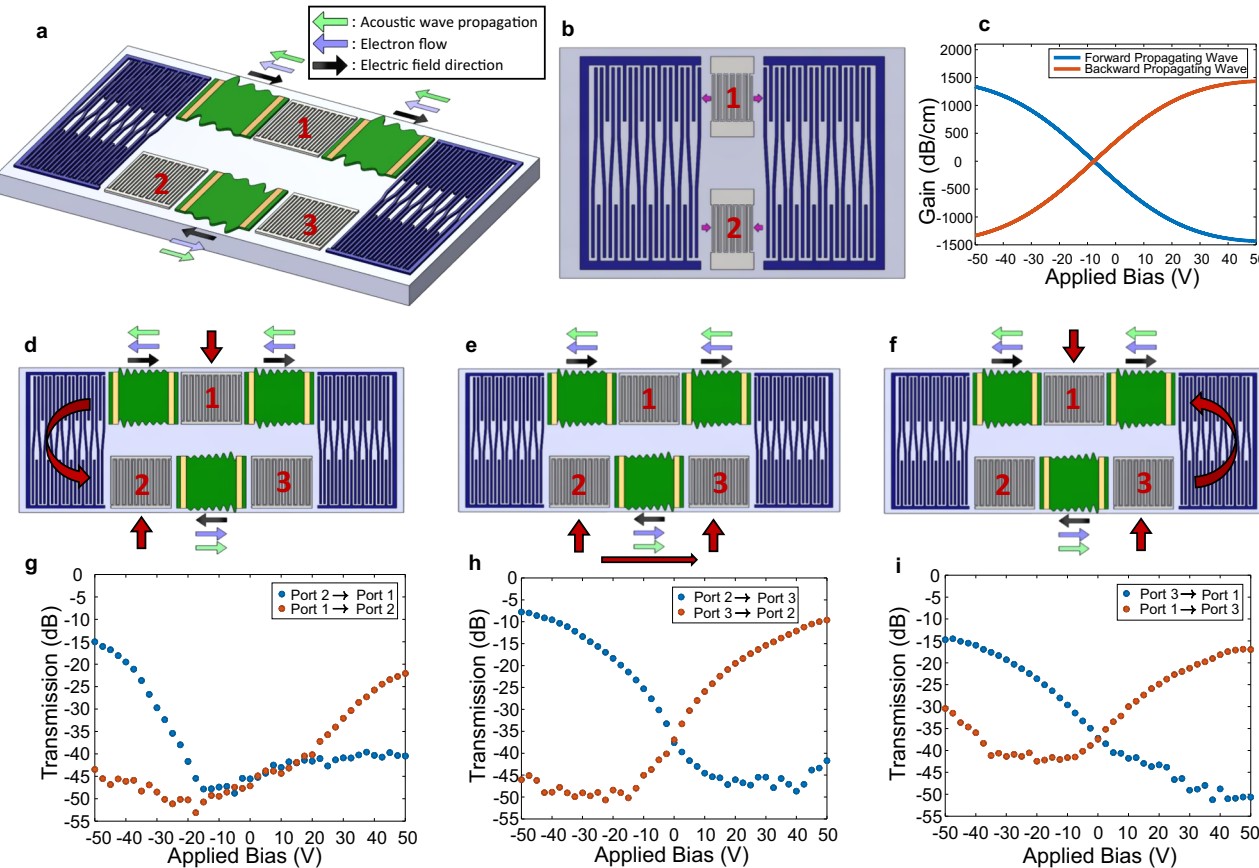

**Fig. 5 Acoustoelectric circulator.** Schematics of the (**a**) three-port acoustoelectric circulator and (**b**) two-port ring filter that is the baseline for the device design. The In$_{0.53}$Ga$_{0.47}$As is shown in green, the LiNbO$_3$ substrate in light gray, the RF IDTs in dark gray, the DC contacts in gold, and the couplers in purple. The light green, light purple, and black arrows denote the directions for the acoustic wave propagation, electron flow, and electric field direction, respectively, for the counterclockwise helicity. **c** Theoretical nonreciprocal gain as a function of applied bias for each acoustoelectric section. Schematics of the measurements done from (**d**) port 1 to port 2, (**e**) port 2 to port 3, and (**f**) port 3 to port 1 with the corresponding experimental data of acoustic transmission as a function of an applied pulsed DC bias shown in (**g**), (**h**), and (**i**), respectively. The acoustic transmission is measured by detecting a change in the S-parameters with respect to time during a continuous RF input at 276 MHz and the application of a 1 ms voltage pulse. A measurement is made every 50 ms giving a duty cycle of 2%.

**Table 1 Circulation insertion loss and isolation for counter-clockwise and clockwise helicities.**

| Ports | Insertion loss (dB) | Isolation (dB) | Helicity |
|---|---|---|---|
| 1 → 2 | 22 | 41 | Counter-clockwise |
| 2 → 3 | 8 | 46 | Counter-clockwise |
| 3 → 1 | 15 | 30 | Counter-clockwise |
| 1 → 3 | 17 | 51 | Clockwise |
| 3 → 2 | 10 | 42 | Clockwise |
| 2 → 1 | 15 | 44 | Clockwise |

Figure 5d–f shows each pair of ports characterized sequentially beginning with ports 1 and 2 (Fig. 5d), then ports 2 and 3 (Fig. 5e), followed by ports 3 and 1 (Fig. 5f). The measured S-parameters for ports 1 and 2, ports 2 and 3, and ports 3 and 1 are shown in Fig. 5g–i, respectively, as a function of the applied bias. The S-parameters are measured with respect to time during the application of a continuous RF signal at 276 MHz and a voltage pulse of 1 ms. The transmission plotted in Fig. 5g–i, is then the S-parameter values measured when the voltage pulse is on. As expected, the experimental contrast ratio increases with increasing applied bias, leading to smaller insertion loss and larger isolation. A summary of the acoustoelectric circulator performance for counter-clockwise and clockwise helicities is given in Table 1. The insertion loss and isolation are the S-parameter values taken from Fig. 5(g–i) at an applied bias of ± 50V. From port 2 to port 3, an insertion loss of 8 dB is obtained with an isolation of 46 dB. While this is the highest performing pair of ports, it is expected that the other pairs of ports will show improved performance with optimization of the RMSCs.

We have demonstrated a proof-of-concept acoustoelectric circulator where the functionality of the device can be modified, depending on the application, by changes to the ports and routing, enabling a novel class of nonreciprocal acoustic wave devices. While our acoustoelectric circulator provides a significantly improved device footprint and a better path forward for on-chip integration compared to commercial circulators that require ferrite-based magnetic materials, further improvements are required to reduce insertion losses, reduce DC power dissipation, enable continuous operation, and improve RF power handling and device linearity. The data shown in Fig. 5 is taken with a continuous RF power input, but a pulsed DC bias. As discussed above in the context of the amplifier, recently we have found that our acoustoelectric devices can operate with a continuously applied DC bias through improved thermal management with a LiNbO$_3$ on Si substrate due to the 30X larger thermal conductivity of Si compared to LiNbO$_8$ (see Supplementary Note 6 and Supplementary Fig. 5). While the linearity of modulated circulator schemes depends primarily on the RF switch size[47], for our acoustoelectric circulator the linearity depends on the RF power handling of the device and the onset of signal compression, which is fundamentally limited by the semiconductor carrier concentration. Therefore, the improved heat dissipation with a LiNbO$_3$ on Si substrate also increases the RF power handling, and therefore the device linearity and dynamic range (see Supplementary Note 6 and Supplementary Fig. 5). Insertion loss can be improved from ports 1 to 2 and ports 3 to 1 by optimization of the RMSC design (see Supplementary Note 7 and Supplementary Fig. 6 for the experimental S-parameters for a metal-only three-port ring filter). Insertion losses can also be significantly improved by exploring different geometries and material stacks of the DC contacts to reduce acoustic reflections. In addition, improvements to the acousto-electric performance through thermal management and further reducing $\sigma t$ will also further optimize the circulator.

## Discussion

Here we have demonstrated the fabrication of passive acoustic delay lines on the same chip as the highest-performing acoustic wave amplifiers ever demonstrated and the first-ever demonstration of an acoustoelectric circulator. This was enabled through our heterogeneously integrated material platform of a 50 nm epitaxial In$_{0.53}$Ga$_{0.47}$As semiconductor with low carrier concentration and large mobility on a 41° YX LiNbO$_3$ piezoelectric substrate, supporting an SH-SAW mode with a $K^2$ of 17%. We have demonstrated SH-SAW acoustic delay line filters with an insertion loss of 8.4 dB and fractional bandwidth of 4.1%, acoustic amplification with 13.5 dB of terminal gain with 15 mW of dissipated DC power, and an acoustoelectric circulator with an isolation of 46 dB.

These experimental demonstrations suggest a promising path forward to develop an all-acoustic and therefore ultra-compact RFSP utilizing the acoustoelectric effect. The fundamental components of a conventional simultaneous transmit and receive (STAR) RFSP for a single frequency band are shown in Fig. 6a. The circulator is implemented in a ferrite material; passive filtering is done with piezoelectric acoustic wave technology; and the power amplifier (PA), low noise amplifier (LNA), local oscillator (LO), and mixer are implemented with CMOS and compound semiconductor electronics. Many modern RF devices, such as Long Term Evolution (LTE) smartphones, require separate RFSPs for more than 30 different frequency bands. As each band needs at least two filters, two mixers, one LO, one PA, and one LNA, this means that more than 60 filters and mixers are required in addition to more than 30 LOs, PAs, and LNAs[16].

Figure 6b shows a concept for the RFSP of Fig. 6a, entirely in the acoustic wave domain. Passive delay line and filtering components are achieved through interactions with periodic electrodes and RMSCs that serve as track changers[44]. All active, nonlinear, and nonreciprocal functions are provided by the acoustoelectric interaction wherever semiconductor material remains after processing. In particular, the LNA and PA are achieved through the addition of acoustic amplification, while the mixer is achieved by the enhanced nonlinearity from electron-phonon coupling, which has previously been applied for convolution[48–50] and correlation[51–53]. The combination of RMSC track changers for routing with the acoustoelectric effect enables an acoustic circulator and LO, as an LO can be viewed as a circulator in which the gain exceeds the round-trip acoustic losses, inducing self-oscillation at the frequency of highest gain. At 276 MHz, the approximate size of the RF acoustics chip in Fig. 6b would be 4.5 mm × 1 mm. For the common LTE band occurring at approximately 1.9 GHz, the smaller acoustic wavelength reduces this size to approximately 0.65 mm × 0.15 mm. In addition to a drastically reduced footprint and improved on-chip device integration, with an all-acoustic platform, there is an increased opportunity to implement signal processing functions passively through the design of the IDT transfer function, such as dispersion control and correlation. This could reduce the required functionality and power requirements of analog-to-digital converters and digital signal processors for applications that are constrained by cost, power, and size requirements. Therefore, this approach could also lead to better device performance and reduced power consumption in addition to the significantly reduced footprint and improved on-chip integration.

Regarding the goal of achieving an all-acoustic, STAR RF front-end, we have experimentally demonstrated three critical components: passive filtering, a high gain amplifier that could be implemented as an LNA, and an acoustic wave circulator. The remaining necessary components are a PA, LO, and frequency mixer. While not achieved in this work, their development should be extensible functionalities of the devices demonstrated here.

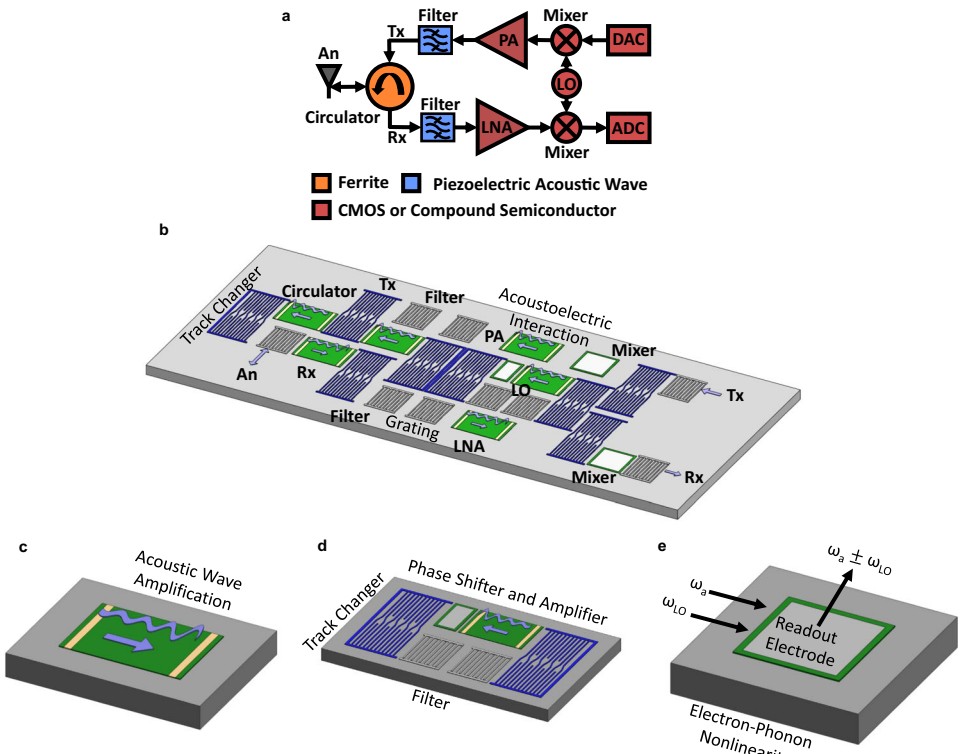

**Fig. 6 RFSP overview and components of an all-acoustic RFSP. a** Fundamental components of a multi-chip approach to an RF front end capable of STAR. The Tx and Rx paths are connected to the antenna (An) through a circulator. The Tx path requires a digital-to-analog converter (DAC) while the Rx path requires an analog-to-digital converter (ADC). **b** Schematic of an all-acoustic RF front end. The $In_{0.53}Ga_{0.47}As$ is shown in green, the $LiNbO_3$ substrate in light gray, DC contacts in gold, aluminum filters in dark gray, and track changers in dark purple. Arrows in light purple indicate the direction of acoustic wave propagation. **c** A PA utilizes the acoustoelectric effect for acoustic wave amplification in a device design optimized for large RF power handling. **d** The LO is constructed from RMSCs, an acoustic filter, an acoustic amplifier, and an acoustic wave phase shifter. **e** An acoustic wave mixer could operate by the nonlinearity from the electron–phonon coupling.

A schematic of a PA is shown in Fig. 6c. While the structure is essentially the same as the acoustic amplifier demonstrated in this work, a PA will require modifications for increased RF power handling. This will likely require both improved thermal dissipation through device design and the addition of an epitaxial layer with increased carrier concentration, $N$, to increase the RF saturation power[39]. In addition, carrier concentration could be varied for different acoustoelectric devices on the same wafer by the inclusion of gate electrodes that control the local carrier concentration with a bias field perpendicular to the acoustic wave propagation.

A notional LO, as shown in Fig. 6d, could be implemented through the inclusion of an acoustic amplifier in a two-port ring filter optimized for low insertion loss, such that the round-trip gain exceeds the round-trip loss. A LO is then similar in function to the circulator demonstrated here and simply requires modification of the track changer path. As described above, gate electrodes could be included to form a phase shifter as a part of the LO. Due to the acoustoelectric interaction, tuning the carrier concentration, $N$, via application of a gate voltage leads to a change in the SAW velocity, which results in a phase shift.

Finally, a mixer, as shown in Fig. 6e, could be developed by extending previous acoustoelectric correlator demonstrations that utilize nonlinear frequency mixing[51–53]. Two SAWs, one at the RF acoustic frequency $\omega_a$ and the other at the LO frequency $\omega_{LO}$, co-propagate along the surface of the acoustoelectric heterostructure. Frequency mixing between the two SAWs occurs through the nonlinear electron-phonon interaction[48,54,55]. A

readout electrode could then be designed and implemented to pull out the desired signal at the upconverted or downconverted frequency.

Potential additional applications of this material platform include acoustic switch networks and the amplification of phonons in combination with $LiNbO_3$ piezoelectric optomechanical devices[56,57]. Devices of this kind could also be utilized in future chip-scale systems for radiofrequency signal processing that use Brillouin interactions in integrated photonics. Given that the gain in Brillouin amplifiers and lasers is inversely proportional to phonon loss rates, the ability to actively control these (and reduce them to zero) could lead to unprecedented performance and novel functionality in these systems[58]. Finally, because the performance of these compound semiconductor devices is expected to improve at cryogenic temperatures[59], these acoustoelectric devices may be able to play the role of circulators and low-noise, high-gain amplifiers for quantum phononic devices[60–62], perhaps playing a role analogous to the Josephson parametric amplifier[63], which has been used for the readout of single microwave photons in superconducting circuit-based quantum computing.

### Data availability

The datasets generated during and analyzed during this study are available from the corresponding author on reasonable request.

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

## Acknowledgements

This work was supported by the Laboratory Directed Research and Development program at Sandia National Laboratories, a multimission laboratory managed and operated by National Technology and Engineering Solutions of Sandia LLC, a wholly-owned subsidiary of Honeywell International Inc. for the U.S. Department of Energy's National Nuclear Security Administration under contract DE-NA0003525. M.E. performed this work, in part, at the Center for Integrated Nanotechnologies, an Office of Science User Facility operated for the U.S. Department of Energy Office of Science. This paper

describes objective technical results and analysis. Any subjective views or opinions that might be expressed in the paper do not necessarily represent the views of the U.S. Department of Energy or the United States Government.

## Author contributions

L.H. and M.E. came up with the device concepts and experimental implementations. L.H., M.M., D.D., A.T., T.F., and M.E. designed the devices and fabrication process flow. M.M., G.P., and S.A. fabricated the devices. L.H. and F.B. performed the measurements. L.H. and M.E. analyzed all data. All authors have given approval to the final version of the manuscript.

## Competing interests

The authors declare no competing interests.
