## [Peer Review File · Nature Communications]

Reviewers' Comments:

Reviewer #1:

This article talks about the vision to integrate all radio-frequency signal processing components on a piezoelectric chip. This target is very promising in the future and is very worthwhile to investigate deeply now. The authors heterogeneously integrate InGaAs CMOS semiconductor on piezoelectric lithium niobate substrate consisting surface acoustic wave filters. The modulation by the acoustic waves on the electron carriers in the InGaAs semiconductor (i.e. acoustoelectric electron-phonon interactions) enlarges the power outputs. Based on this principle, they combine passive filters with time delay, amplifiers and non-reciprocal circulators on one single chip.

However, in their previous publication, Hackett, L, et al., “High-gain leaky surface acoustic wave amplifier in epitaxial InGaAs on lithium niobate heterostructure”, *Appl. Phys. Lett.* 114, 2019, the concept of acoustic induced electron amplifier on one single chip has been presented already. Thus, the concept is not novel to me. Although this manuscript displays the first-ever nonmagnetic acoustoelectric non-reciprocal circulators, however the performance is worse than commercial products. Besides, the remaining necessary components such as a local oscillator, low noise amplifier, and frequency mixer, there is no experimental results. The authors just give some proof of concept descriptions. thus I do not think it is a full research and do not suitably for nature communication journal.

Reviewer #2:

Remarks to the Author:

This is an excellent and original contribution. Over the last few years acousto-electric effect has been explored by several groups including my own. However almost all except one effort has only been able to show "Less Loss" and no AE Gain. This research paper from M. Eichenfield's team at Sandia is the first group to my knowledge to demonstrate Gain using conventional materials.

It is the opinion of this reviewer that the Initial motivation of STAR radios should be down-played. The AE-Gain demonstrated is for extremely low RF signal powers, and only in Pulsed mode operation. such a Pulsed or gated mode of operation (maximum 50% shown) directly takes away from the 2X increase in radio bandwidth touted by the authors for STAR?

The RF input power Dynamic Range is small. the AE-effect gain amplifiers need to outperform the best-in-class GaAs or GaN amplifiers. With such a low and small RF input power, its difficult for this reviewer to imagine the amplifier being used in any Radio.

In both these instances, the scientific and engineering innovation are to be lauded, but without a clearly articulated and analytically calculated scientific path forward, this reviewer is not convinced AE is the way forward as the authors motivate it out to be. In a way, the authors are overselling the technology and the reviewer thinks this does disservice to the fundamental achievements in the paper, which are amazing.

In the case of the AE-circulator, the reviewer could not figure out if the measurements were performed in RF-pulsed mode of CW-mode? Can the authors please include S11, S21, S32 and S31 measurements of the circulator. The reviewer requests the authors to include non-linearity data (or excellent linearity data) and compare that with demonstrated modulated circulator schemes.

Reviewer #3:

Remarks to the Author:

This paper presents the design of acoustoelectric delay lines, amplifiers, and filters. The most interesting contribution is the circulator, as it is known to be a difficult block to build in a small area or volume, so this, with better optimization to improve insertion loss, can potentially be a big breakthrough.

Regarding the amplifier, I don't see any gain anywhere. The plots in the supplementary material show a negative gain in dB. How can this be called an amplifier? Please clearly show an S21 plot indicating positive gain. Also, when describing gain, please be sure to very clearly indicate what type of gain is being described (voltage gain? power gain? what are the terminal impedances?).

How are the devices matched to 50 ohms (or are they?), presumably to enable testing with standard RF equipment?

I suggest leaving the discussion about an all-acoustic receiver to the discussion section, not the main part of the text, as this has not been demonstrated in the manuscript. In addition, please better justify what the advantages of such an approach would be, given the current (major) limitations of the proposed technology. This will give the reader better context to understand how the future of this technology will look like.

The thermal run away conditions appear to be a major challenge as well, preventing normal operation. It would be nice to see additional discussion regarding this, and what could be done to address this in the future.

Reviewer 2:

1. “This is an excellent and original contribution. Over the last few years acousto-electric effect has been explored by several groups including my own. However almost all except one effort has only been able to show "Less Loss" and no AE Gain. This research paper from M. Eichenfield's team at Sandia is the first group to my knowledge to demonstrate Gain using conventional materials.”

Answer to the reviewer:

We appreciate the recognition from the reviewer of our experimental accomplishments, who as stated works specifically in the technical area of this manuscript.

2. “It is the opinion of this reviewer that the Initial motivation of STAR radios should be down-played. The AE-Gain demonstrated is for extremely low RF signal powers, and only in Pulsed mode operation. such a Pulsed or gated mode of operation (maximum 50% shown) directly takes away from the 2X increase in radio bandwidth touted by the authors for STAR?”

Answer to the reviewer:

We agree with the reviewer that the emphasis of the manuscript should be to highlight our key experimental results and therefore we have moved our presentation of an all-acoustic radiofrequency signal processor to the Discussion section. In addition, we agree with the reviewer concerns about the future direction of acoustoelectric device technology given the low radiofrequency signal powers and pulsed mode operation. To address these issues we have included recent experimental results in the Supplementary Information on continuous operation of an acoustoelectric amplifier with larger RF power handling using the same device structures and principals, but replacing the lithium niobate substrate with a lithium niobate on silicon substrate, which improves the thermal management through the improved thermal conductivity of silicon as opposed to lithium niobate. Our preliminary results show 15.4 dB of electronic gain in a 250 μm long device operating with a continuously applied DC bias of 45 volts. We also demonstrate a radiofrequency output power of 8.7 dBm.

The Introduction and Discussion Sections have been restructured such that our presentation of an all-acoustic radiofrequency signal processor using the acoustoelectric effect is moved to the Discussion Section. These sections are not copy and pasted here, but can be found in the revised manuscript.

Acoustic Amplifier Section:

“The critical obstacles to integrate acoustoelectric devices into future RFSPs are stable operation with a continuously applied DC bias in addition to increasing the RF input power dynamic range and overall RF power handling. We have recently found that by using a LiNbO_3 film on Si instead of a bulk LiNbO_3 substrate we can support a guided acoustic wave and improve the heat dissipation efficiency through a 30X improvement of the thermal conductivity of Si compared to LiNbO_3 .^{40,41} The device fabrication remains identical except for replacing the LiNbO_3 substrate with a LiNbO_3 on Si substrate. In a 250 μm long device, we achieve an electronic gain of 15.4 dB while operating continuously and a RF output power of 8.7 dBm (see Supplementary Note 7 and Supplementary Fig. 6). These initial results suggest that the LiNbO_3 film on bulk Si substrate is a promising path forward to overcome the challenges of thermal management and power handling for active acoustic wave devices and the material and device parameters will be further optimized in our future work.”

Supplementary Information:

“Supplementary Note 6. Acoustic wave amplifier operating continuously on a LiNbO_3 on Si substrate

The potential to improve the thermal management of active acoustic wave devices in our material platform was assessed by replacing the 41°YX LiNbO_3 substrate with a substrate consisting of 5 μm thick film of YX LiNbO_3 on bulk Si. The difference in acoustic impedance between Si and LiNbO_3 causes a launched acoustic wave to be guided in the LiNbO_3 film. Si has a thermal conductivity of 150 W/m-K while LiNbO_3 has a thermal conductivity of 4.6 W/m-K.^{4,5} Therefore, it is expected that heat dissipation will be significantly improved through the 30X increase in thermal conductivity for Si over LiNbO_3 . Supplementary Figure 5(a) shows the measured S_{21} as a function of frequency with increasing applied bias for a 250 μm long device with a 75 nm thick $\text{In}_{0.53}\text{Ga}_{0.47}\text{As}$ amplifier layer with a measured Hall mobility of 2360 $\text{cm}^2/\text{V}\cdot\text{s}$ and a doping concentration of $1.9 \times 10^{16} \text{ cm}^{-3}$. The device shows stable operation with a continuously applied DC bias. Supplementary Figure 5(b) shows the electronic gain as a function of applied bias. An electronic gain of 15.4 dB is achieved with an applied

bias of 45 V. As can be seen, there is a corresponding attenuation of 13.6 dB for the backward propagating wave. Although this device did not have as high of a gain performance as the devices presented in the main text, this is attributed to differences in the epitaxial layer that we can amend in the future and thus have both thermal dissipation advantages and exceptional gain performance in the same device. Supplementary Figure 5(c) shows the RF input power as a function of the RF output power. The device shows stable performance for the range of tested RF input powers, up to -1.6 dBm, and a linear response up to a RF output power of 6 dBm.

Supplementary Figure 5. (a) Measured S_{21} as a function of frequency with increasing applied bias for a 75 nm thick and 250 μm long $\text{In}_{0.53}\text{Ga}_{0.47}\text{As}$ layer on a 5 μm YX LiNbO_3 layer on a bulk Si substrate. (b) Electronic gain as a function of applied bias. (c) RF output power as a function of RF input power.”

3. “The RF input power Dynamic Range is small. the AE-effect gain amplifiers need to outperform the best-in-class GaAs or GaN amplifiers. With such a low and small RF input power, its difficult for this reviewer to imagine the amplifier being used in any Radio.”

Answer to the reviewer:

We agree with the reviewer that further extension of the acoustoelectric device technology requires a larger radiofrequency input power dynamic range. Therefore, we have included recent experimental results in the Supplementary Information showing a larger radiofrequency input power dynamic range using a lithium niobate on silicon substrate, which improves power handling through improved thermal management due to the improved thermal conductivity of silicon as opposed to lithium niobate. The 250 μm long device shows stable performance for the range of tested radiofrequency input powers, up to -1.6 dBm, and a linear response up to a radiofrequency output power of 6 dBm. We expect further improvements in device design and fabrication will result in even higher dynamic ranges.

Overall this is still a relatively recent technology. We agree that the devices do not outperform the best-in-class GaAs or GaN amplifiers. However, there may be a potential for acoustoelectric devices to out-perform best-in-class amplifiers with further device and materials optimization. In addition, for specific applications where size is paramount, there is a potential appeal with the size reduction that comes inherently with an all-acoustic approach.

See above modifications to the Acoustic Amplifier Section and the Supplementary Information (response to comment 2 from Reviewer 2).

4. “In both these instances, the scientific and engineering innovation are to be lauded, but without a clearly articulated and analytically calculated scientific path forward, this reviewer is not convinced AE is the way forward as the authors motivate it out to be. In a way, the authors are overselling the technology and the reviewer thinks this does disservice to the fundamental achievements in the paper, which are amazing.”

Answer to the reviewer:

We appreciate the recognition from the reviewer that the “scientific and engineering innovation are to be lauded” and that the “fundamental achievements in the paper” are “amazing.” We have modified the manuscript to emphasize these accomplishments and we have moved our presentation of an all-acoustic radiofrequency signal processor to the Discussion. We hope that with the added experimental results to the Supplementary Information of the manuscript showing continuous operation, improved radiofrequency power handling, and improved dynamic range, convince the reviewer that acoustoelectric technology is a compelling path forward for a future all-acoustic and therefore ultra-compact radiofrequency signal processor. However, we agree with the reviewer that there remain significant technological hurdles, which is why we have moved this section to the Discussion and instead emphasize our experimental results.

See above modifications to the Acoustic Amplifier Section and the Supplementary Information (response to comment 2 from Reviewer 2).

The Introduction and Discussion Sections have been restructured such that our presentation of an all-acoustic radiofrequency signal processor using the acoustoelectric effect is moved to the Discussion Section. These sections are not copy and pasted here, but can be found in the revised manuscript.

5. “In the case of the AE-circulator, the reviewer could not figure out if the measurements were performed in RF-pulsed mode or CW-mode? Can the authors please include S11, S21, S32 and S31 measurements of the circulator. The reviewer requests the authors to include non-linearity data (or excellent linearity data) and compare that with demonstrated modulated circulator schemes.”

Answer to the reviewer:

We appreciate the additional clarification and data requests on the acoustoelectric circulator. The circulator measurements are performed with a continuous radiofrequency input and a pulsed DC bias. This has been clarified in the manuscript. We have added the requested S-parameter data to the Supplementary Information. The linearity of modulated circulator schemes depends primarily on the RF switch size. For our acoustoelectric circulator, the linearity depends on the radiofrequency power handling of the device and the onset of signal compression, which is fundamentally limited by the semiconductor carrier concentration. We have added this discussion to the Acoustoelectric Circulator section of our manuscript.

Acoustoelectric Circulator Section:

“While our acoustoelectric circulator provides a significantly improved device footprint and a better path forward for on-chip integration compared to commercial circulators that require ferrite-based magnetic materials, further improvements are required to reduce insertion losses, reduce DC power dissipation, enable continuous operation, and improve RF power handling and device linearity. The data shown in Fig. 5 is taken with a continuous RF power input, but a pulsed DC bias. As discussed previously, recently we have found that our acoustoelectric devices can operate with a continuously applied DC bias through improved thermal management with a LiNbO₃ on Si substrate due to the 30X larger thermal conductivity of Si compared to LiNbO₃ (see Supplementary Note 6 and Supplementary Figure 5). While the linearity of modulated circulator schemes depends primarily on the RF switch

size,⁴⁵ for our acoustoelectric circulator the linearity depends on the RF power handling of the device and the onset of signal compression, which is fundamentally limited by the semiconductor carrier concentration. Therefore, the improved heat dissipation with a LiNbO₃ on Si substrate also increases the RF power handling and therefore the device linearity and dynamic range (see Supplementary Note 6 and Supplementary Figure 5).”

Supplementary Information:

Supplementary Figure 6. Measured (a) S_{11} , (b) S_{21} , (c) S_{32} , and (d) S_{31} of the acoustoelectric circulator.

Reviewer 3:

1. “This paper presents the design of acoustoelectric delay lines, amplifiers, and filters. The most interesting contribution is the circulator, as it is known to be a difficult block to build in a small area or volume, so this, with better optimization to improve insertion loss, can potentially be a big breakthrough.”

Answer to the reviewer:

We agree with the reviewer that the experimental demonstration of the first-ever acoustoelectric circulator is a significant and important contribution made by this manuscript.

2. “Regarding the amplifier, I don't see any gain anywhere. The plots in the supplementary material show a negative gain in dB. How can this be called an amplifier? Please clearly show an S21 plot indicating positive gain. Also, when describing gain, please be sure to very clearly indicate what type of gain is being described (voltage gain? power gain? what are the terminal impedances?).”

Answer to the reviewer:

The amplifier gain is shown in the manuscript in Figures 4(b) and 4(c) for electronic gain in units of dB/cm and terminal gain in units of dB, respectively. In order to achieve electronic gain the amplification must be large enough to overcome losses from the acoustoelectric effect while to achieve

terminal gain the amplification must be large enough to overcome all of the losses in the system, including transducer conversion losses and losses from reflections due to the thick DC contact metal. **In this manuscript we demonstrate an electronic power gain exceeding 800 dB/cm and a terminal gain of 13.5 dB in a device only 505 μm long.** The terminal gain plot (Figure 4(c)) is therefore exactly a S21 plot indicating net positive gain. The experimental data is for power gain matched to a source impedance of 50 Ω .

Supplementary Figure 4, which shows that the gain bandwidth is equivalent to the transducer bandwidth, shows gain because we see a significant increase in S21 with an applied bias. However, the gain is not large enough to overcome all losses, primarily introduced from the transducers and DC contacts, and therefore terminal gain is not achieved and the overall S21 is negative. This device did not achieve terminal gain, but was used to demonstrate that the gain bandwidth is equal to the transducer bandwidth. Terminal gain was not achieved in this particular device primarily because it had a thicker amplifier layer of 300 nm, which reduces the gain slope. To avoid confusion, we removed the figure from the Supplementary Information.

We have added additional details describing our experimentally demonstrated gain to the Acoustic Amplifier Section:

Acoustic Amplifier Section:

“Here we present our experimental gain data in two ways. One is the electronic gain, which can only be achieved if the amplification is large enough to overcome losses due to the acoustoelectric effect. The other type of gain we present is terminal gain, which can only be achieved once the amplification is large enough to overcome all of the losses in the system, including transducer conversion losses and reflection losses off of the thick DC contact metal. Both electronic gain and terminal gain are power gain measured on a network analyzer with a source impedance of 50 Ω .”

3. “How are the devices matched to 50 ohms (or are they?), presumably to enable testing with standard RF equipment?”

Answer to the reviewer:

The devices are matched to 50 Ω through a two-step process. First the Mason equivalent circuit model is used in order to approximate the number of electrode pairs required to match to 50 Ω given the device geometry, expected capacitance, electromechanical properties, and operating frequency. This is then experimentally verified by fabricating an array of delay line devices with a varying number of electrode pairs. The S11 and S22 are measured on a network analyzer as a function of the number of electrode pairs to find the location of optimum impedance matching. When the semiconductor material is added, there will likely be a small modification to the impedance matching condition. Therefore for the acoustoelectric devices we are likely close to matching to 50 Ω , but we must and do consider impedance mismatch losses when determining important experimental parameters such as the radiofrequency input power to the device.

We have added a description about matching to 50 Ω to the Acoustic Delay Line section of the manuscript:

Acoustic Delay Line Section:

“For the delay lines on YZ and 41° YX LiNbO₃, matching to the 50 Ω source impedance of the network analyzer was optimized through a two-step process. The approximate number of electrode pairs was calculated using the Mason equivalent circuit model given the IDT aperture, expected capacitance, K^2 , and operating frequency.³⁶ A bank of delay lines was then fabricated with a varying

number of electrode pairs followed by measuring the S-parameters as a function of the number of electrode pairs to determine the optimal number.”

4. “I suggest leaving the discussion about an all-acoustic receiver to the discussion section, not the main part of the text, as this has not been demonstrated in the manuscript. In addition, please better justify what the advantages of such an approach would be, given the current (major) limitations of the proposed technology. This will give the reader better context to understand how the future of this technology will look like.”

Answer to the reviewer:

We agree with the reviewer that the presentation of an all-acoustic radiofrequency signal processor using the acoustoelectric effect should be moved to the Discussion section of the manuscript and that the advantages and limitations of the approach should be discussed in more detail. The advantages of an all-acoustic radiofrequency signal processor are primarily a reduced footprint (by several orders of magnitude) and improved on-chip integration. In addition, in an all-acoustic platform there is an increased opportunity to implement signal processing functions passively through the design of the interdigital transducer transfer function such as dispersion control and correlation. This could reduce the required functionality and power requirements of analog-to-digital converters and digital signal processors for applications that are constrained by cost, power, and size requirements. Therefore, our vision could also lead to better device performance and reduced power consumption in addition to the significantly reduced footprint and improved on-chip integration.

The Introduction and Discussion Sections have been restructured such that our presentation of an all-acoustic radiofrequency signal processor using the acoustoelectric effect is moved to the Discussion Section. These sections are not copy and pasted here, but can be found in the revised manuscript.

5. “The thermal run away conditions appear to be a major challenge as well, preventing normal operation. It would be nice to see additional discussion regarding this, and what could be done to address this in the future.”

Answer to the reviewer:

We agree with the reviewer that proper thermal management in order to reduce thermal drift and run-away and enable continuous operation are important and requires further discussion in the manuscript. We believe that the best path forward is to replace the lithium niobate substrate with a lithium niobate thin film on silicon substrate to take advantage of the improved thermal conductivity of silicon over lithium niobate. We have included our preliminary experimental results using this substrate in the Supplementary Information where we achieve 15.4 dB of electronic gain in a 250 μm long device with a continuously applied DC bias of 45 volts.

See above modifications to the Acoustic Amplifier Section and the Supplementary Information (response to comment 2 from Reviewer 2).

Reviewer 1:

1. “This article talks about the vision to integrate all radio-frequency signal processing components on a piezoelectric chip. This target is very promising in the future and is very worthwhile to investigate deeply now. The authors heterogeneously integrate InGaAs CMOS semiconductor on piezoelectric lithium niobate substrate consisting surface acoustic wave filters. The modulation by the acoustic waves on the electron carriers in the InGaAs semiconductor (i.e. acoustoelectric electron-phonon

interactions) enlarges the power outputs. Based on this principle, they combine passive filters with time delay, amplifiers and non-reciprocal circulators on one single chip.”

Answer to the reviewer:

We appreciate the acknowledgement from the reviewer that the development of an all-acoustic radiofrequency signal processor is of significant interest to a larger scientific community and promising approach for future radiofrequency technologies.

2. “However, in their previous publication, Hackett, L, et al., “High-gain leaky surface acoustic wave amplifier in epitaxial InGaAs on lithium niobate heterostructure”, *Appl. Phys. Lett.* 114, 2019, the concept of acoustic induced electron amplifier on one single chip has been presented already. Thus, the concept is not novel to me.”

Answer to the reviewer:

We never suggest that the novelty of our manuscript is to demonstrate a monolithic acoustic wave amplifier. In fact, this has been done by others as we have referenced in the manuscript. However, what we do demonstrate in this manuscript is (1) the best performing acoustoelectric amplifier ever, including drastic improvements in both gain and power consumption over our previous work in this area, (2) the first-ever demonstration of an acoustoelectric circulator, and (3) the co-fabrication of these components with high performing passive acoustic wave filters. Our experimental breakthroughs then allow us to propose a new vision for an all-acoustic radiofrequency signal processor based on the acoustoelectric effect.

3. “Although this manuscript displays the first-ever nonmagnetic acoustoelectric non-reciprocal circulators, however the performance is worse than commercial products.”

Answer to the reviewer:

As this is the first-ever demonstration, we do not seek to compete directly with commercial magnetic circulators that that have been developed for over 50 years. We understand that this particular implementation doesn't match the performance characteristics of magnetic circulators, but there is significant room for improvement to further advance acoustoelectric circulator technology. In addition, there are multiple metrics to which these devices should be compared. If you were to pick the size, then our acoustoelectric circulator is reduced in area by greater than 1000X compared to a commercially available radiofrequency circulator. We have added additional discussion on the path to make acoustoelectric circulators a competitive commercial technology in the future to the Acoustoelectric Circulator section of the paper.

See above modifications to the Acoustoelectric Circulator Section (response to comment 5 from Reviewer 2).

4. “Besides, the remaining necessary components such as a local oscillator, low noise amplifier, and frequency mixer, there is no experimental results. The authors just give some proof of concept descriptions. thus I do not think it is a full research and do not suitably for nature communication journal.”

Answer to the reviewer:

We believe that the experimental results we provide in the manuscript and the Supplemental Information are extensive and show a breakthrough in acoustoelectric device technology that completes a full research article. The reviewer comment that we have not included experimental results for a low noise amplifier is incorrect as we have included an entire section on the demonstration of an acoustic amplifier with large terminal gain. As discussed in the paper, a local oscillator is similar in function to a circulator, which is experimentally demonstrated, and simply requires modification of the track changer path. Also discussed in the paper is that a frequency mixer is an extension of a high

performing acoustic wave amplifier. Overall, we believe the experimental work we have demonstrated to constitute an original and complete manuscript with significant impact for the future of acoustoelectric device technology as applied to radiofrequency signal processing.

Sincerely,

Matt Eichenfield, Ph.D.
Distinguished Member of Technical Staff
Sandia National Laboratories

Reviewers' Comments:

Reviewer #1:

Remarks to the Author:

The revised version of the manuscript is more solid than the original one in terms of experimental data. The authors put the promising vision in the discussion part and the flow of the work is more clearly and reasonable in the revised version. The authors emphasize outcomes of highest performing surface acoustic wave amplifiers and the first ever nonmagnetic acoustoelectric circulator. However I do not think there are breakthrough contributions deserving this journal both in science and technology to the society taking into account the integration concept of all component are available for a long time and the overall performance of the device is not outstanding.

Reviewer #2:

Remarks to the Author:

This reviewer enjoyed reading the manuscript turned around in this manner, which emphasizes the measurements of acoustoelectricity in this platform.

AE Circulator:

The reviewer recommends the circulator statement on line 20, also state "pulsed DC" mode of operation as also stated in the rebuttal letter and line 412 of the manuscript.

Furthermore it should be added to Figure 5 of the main text. Infact, in the caption when talking about fig 5.g,h,i the authors are requested to state the exact measurement conditions, what was the DC pulse frequency and % duty cycle? Is the transmission "broadband" in the bandwidth of delay-line element or at a single RF tone? if RF tone, which RF tone?

This reviewer fails to see any broadband circulation in the data provided in Supplementary Figure 6 and referenced in lines 400-402 of the main text. In addition just like stated in Table1 of main text, the Helicity should be stated in the caption of Supplementary Figure 6, and for that helicity, there should be atleast one backward plot so in addition to S31, there should be S13? Either way any markings, annotations of the raw data and how the values in Table1 were extracted from the raw data will be helpful.

Acoustic Wave Amplifier (Supplementary Note 6 line 148) and Main text Figure 4.c):

It was exciting to see continuous DC operation. The reviewer seeks clarity on the definition of Electronic Gain Supplementary Figure 5.b and how it is extracted from Figure 5.a. Is it at a particular frequency? averaged across the bandwidth of the main lobe?

The reviewer thinks the electronic gain is defined as the Insertion Loss at one frequency say 245MHz, at 0V DC bias and then at 45V DC bias and subtracting the two? Is this correct? Either way the authors should elucidate.

Moreover, Figure 5.c says at 0dBm input power, at 45 Volts, there is 9dBm output power. That would be a port to port 9dB gain. Is this correct? Because the S21 plot in 5.a shows "less loss" ?

Reviewer #3:

None

Reviewer #4:

Remarks to the Author:

I am being asked to review a revised manuscript. This is very impressive work representing a

significant technological advance for on-chip RF acoustic signal processing. The authors have addressed comprehensively the comments which has improved the manuscript. I was initially concerned that the performance does not meet the performance of commercial circulators as pointed out by one of the reviewers but my view is that this is fine as a proof of concept as the main advance here, which is heroic, is the heterogeneous integration of components to realize the functionality. If the performance exceeded commercial products this paper would probably be considered for Nature of Nature Electronics. This is an exciting area of research with a vibrant community, particularly with the marriage of photonc integration which should play into this in the long term - can the authors comment?

Reviewer 2:

1. “The reviewer recommends the circulator statement on line 20, also state "pulsed DC" mode of operation as also stated in the rebuttal letter and line 412 of the manuscript.”

Answer to the reviewer:

We agree with the reviewer and have made this modification.

Line 20 of the manuscript has been modified:

“We also demonstrate the first-ever nonmagnetic acoustoelectric circulator with an isolation of 46 dB with a pulsed DC bias.”

2. “Furthermore it should be added to Figure 5 of the main text. In fact, in the caption when talking about fig 5.g,h,i the authors are requested to state the exact measurement conditions, what was the DC pulse frequency and % duty cycle? Is the transmission "broadband" in the bandwidth of delay-line element or at a single RF tone? if RF tone, which RF tone?”

Answer to the reviewer:

We agree with the reviewer that the caption of Figure 5 in the main text should be modified to clarify the exact measurement conditions.

The figure caption for Figure 5 in the main text has been modified:

“Fig. 5: Acoustoelectric circulator. Schematics of the (a) three port acoustoelectric circulator and (b) two port ring filter that is the baseline for the device design. (c) Theoretical nonreciprocal gain as a function of applied bias for each acoustoelectric section. Schematics of the measurements done from (d) port 1 to port 2, (e) port 2 to port 3, and (f) port 3 to port 1 with the corresponding experimental data of acoustic transmission as a function of an applied pulsed DC bias shown in (g), (h), and (i), respectively. The acoustic transmission is measured by detecting a change in the S-parameters with respect to time during a continuous RF input at 276 MHz and the application of a 1 ms voltage pulse. A measurement is made every 50 ms giving a duty cycle of 2%.”

3. “This reviewer fails to see any broadband circulation in the data provided in Supplementary Figure 6 and referenced in lines 400-402 of the main text. In addition just like stated in Table1 of main text, the Helicity should be stated in the caption of Supplementary Figure 6, and for that helicity, there should be at least one backward plot so in addition to S31, there should be S13? Either way any markings, annotations of the raw data and how the values in Table1 were extracted from the raw data will be helpful.”

Answer to the reviewer:

We thank the reviewer for this comment to help us clarify the circulator data. The data is taken by measuring a change in the S-parameters with respect to time during a continuous RF input at a single frequency (276 MHz) and the application of a 1 ms voltage pulse. All of the relevant circulator data is shown in Figure 5 and Table 1 in the main text in addition to Supplementary Figure 7. Therefore we have removed Supplementary Figure 6 and the reference to it in the main text. We agree that the text should be modified to clarify how the values in Table 1 were extracted from the raw data shown in Figure 5 and this has been added to the main text. In addition, we have further clarified the description of the circulator measurements in the main text (see below).

Modified main text description of Figure 5:

“Figures 5(d-f) show each pair of ports characterized sequentially beginning with ports 1 and 2 (Fig. 5(d)), then ports 2 and 3 (Fig. 5(e)), followed by ports 3 and 1 (Fig. 5(f)). The measured S-parameters for ports 1 and 2, ports 2 and 3, and ports 3 and 1 are shown in Figs. 5(g-i), respectively, as a function of the applied bias. The S-parameters are measured with respect to time during the application of a continuous RF signal at 276 MHz and a voltage pulse of 1 ms. The transmission, plotted in Figs. 5(g-i), is then the S-parameter values measured when the voltage pulse is on. As expected, the experimental contrast ratio increases with increasing applied bias, leading to smaller insertion loss and larger isolation. A summary of the acoustoelectric circulator performance for counter-clockwise and clockwise helicities is given in Table 1. The insertion loss and isolation are the S-parameter values taken from Figs. 5(g-i) at an applied bias of $\pm 50V$. From port 2 to port 3, an insertion loss of 8 dB is obtained with an isolation of 46 dB. While this is the highest performing pair of ports, it is expected that the other pairs of ports will show improved performance with optimization of the RMSCs.”

4. “It was exciting to see continuous DC operation. The reviewer seeks clarity on the definition of Electronic Gain Supplementary Figure 5.b and how it is extracted from Figure 5.a. Is it at a particular frequency? averaged across the bandwidth of the main lobe? The reviewer thinks the electronic gain is defined as the Insertion Loss at one frequency say 245MHz, at 0V DC bias and then at 45V DC bias and subtracting the two? Is this correct? Either way the authors should elucidate.”

Answer to the reviewer:

We thank the reviewer for this comment, which will help us to clarify the definition of electronic gain in the manuscript. Electronic gain can only be achieved if the amplification is large enough to overcome losses due to the acoustoelectric effect. To be more specific, acoustic gain only occurs when the electron drift velocity exceeds the acoustic velocity. Acoustic loss occurs when the electron drift velocity is less than the acoustic velocity. Therefore a voltage must be applied to reach 0 dB of electronic gain. We determine this voltage based on the device length and the measured semiconductor mobility. For the device data shown in Supplementary Figure 5, this voltage is 4V. Therefore electronic gain is defined as subtracting the insertion loss at one frequency (in this case 253 MHz) at 4V DC bias and 45V DC bias.

The following was added to Supplementary Note 6. Acoustic wave amplifier operating continuously on a LiNbO₃ on Si substrate in the Supplementary Information:

“Electronic gain occurs in these devices when the amplification is large enough to overcome the losses due to the acoustoelectric effect. Gain only occurs when the electron drift velocity exceeds the acoustic velocity and therefore a voltage must be applied to reach 0 dB of electronic gain. The value of this voltage depends on the device length and the semiconductor mobility. For this device the required applied bias to reach 0 dB of electronic gain is 4V. Therefore, the electronic gain is defined based on the subtraction of the insertion loss at 253 MHz with an applied bias of 4V and 45V.”

5. “Moreover, Figure 5.c says at 0dBm input power, at 45 Volts, there is 9dBm output power. That would be a port to port 9dB gain. Is this correct? Because the S21 plot in 5.a shows "less loss" ?”

Answer to the reviewer:

We appreciate this comment from the reviewer that allows us to clarify the power dependence of the acoustic gain in our acoustoelectric amplifier devices. For Supplementary Figure 5(c), we followed the established method for determining the acoustoelectric amplifier gain compression and saturation power as described in the following reference: Coldren, L. A. & Kino, G. S. Monolithic Acoustic Surface-Wave Amplifier. *Appl Phys Lett* 18, 317, (1971). We estimate the input power by taking into consideration the attenuation with no drift field, which is approximately -33 dB. From the experimental gain curve, we find that 5.6 dB is due to loss associated with the acoustoelectric effect according to the

calculated voltage to reach the 0 dB operating point based on the device length and the measured semiconductor mobility of $2000 \text{ cm}^2/\text{V}\cdot\text{s}$. The remaining 27.4 dB is split between the input and output to account for transducer conversion losses and acoustic losses from the wave traversing the thick DC contacts. The input power is then the source power from the network analyzer minus the losses from the input transducer and DC contact. The output power is the input power plus the electronic gain. We find that our current labeling of the plotted experimental data may cause confusion and therefore have updated the labels to read “Acoustic Power In (dBm)” and “Acoustic Power Out (dBm)”. We describe the power calculations in Supplementary Note 4 and have modified that note to include additional clarifying details.

Supplementary Note 4 has been modified:

“Supplementary Note 4. Acoustic input power dependence and power efficiency

We measured the acoustic output power and power conversion efficiency as a function of the dissipated DC power for different acoustic input powers. Here we estimate the input acoustic power by taking into consideration the attenuation with no drift field, which is approximately -33 dB. From the experimental gain curve, we find that 5.6 dB is due to loss associated with the acoustoelectric effect according to the calculated voltage to reach the 0 dB operating point based on the device length and the measured semiconductor mobility (μ) of $2000 \text{ cm}^2/\text{V}\cdot\text{s}$. The remaining 27.4 dB is split between the input and output to account for transducer conversion losses and losses from the DC contacts. By measuring the insertion loss of acoustic delay lines without and with the DC contact metal stack used here, we have found that loss from reflecting from the DC contacts on account of the large impedance change under the thick, high-density metal can be as high as 5 dB per contact. This added insertion loss could be improved by reducing the thickness of the DC contact metal. Here a thickness exceeding $1 \mu\text{m}$ was used to ensure good Ohmic contact, but since the applied fields are large enough to overcome contact barriers, this is likely not required. The acoustic input power is then the source power with the losses from the input transducer and DC contact while the acoustic power is the input power with the added electronic gain.

A plot of acoustic output power as a function of the dissipated DC power for different acoustic input powers is shown in Supplementary Figure 4(a) for a $505 \mu\text{m}$ long amplifier device. Gain rollover can be seen, as there is less change in acoustic output power with increasing dissipated DC power. This is likely due to thermal effects. In addition, gain compression can also be seen as there is less change in the acoustic output power for increasing acoustic input powers. Supplementary Figure 4(b) shows the power efficiency (η), defined as $\eta = \frac{P_{OUT}^A}{P_{DISS}^{DC}}$, plotted as a function of the dissipated DC power for different acoustic input powers where P_{OUT}^A is the acoustic output power and P_{DISS}^{DC} is the dissipated DC power.”

Modified main text that references Supplementary Note 4:

“We also measured the acoustic output power and DC to RF *acoustic* power efficiency as a function of the DC power dissipation and acoustic input power (see Supplementary Note 4 and Supplementary Fig. 4). The power efficiency (η) is defined as $\eta = \frac{P_{OUT}^A}{P_{DISS}^{DC}}$ where P_{OUT}^A is the acoustic output power and P_{DISS}^{DC} is the dissipated DC power. We achieve an η of 4.4% at an acoustic input power of -34 dBm and a DC power dissipation of 12 mW.”

Modified Supplementary Figure 5 figure caption:

“Supplementary Figure 5. (a) Measured S_{21} as a function of frequency with increasing applied bias for a 75 nm thick and $250 \mu\text{m}$ long $\text{In}_{0.53}\text{Ga}_{0.47}\text{As}$ layer on a $5 \mu\text{m}$ YX LiNbO_3 layer on a bulk Si substrate.

(b) Electronic gain as a function of applied bias. (c) Acoustic output power as a function of acoustic input power.”

Reviewer 4:

1. “This is an exciting area of research with a vibrant community, particularly with the marriage of photonic integration which should play into this in the long term - can the authors comment?”

Answer to the reviewer:

This is an interesting comment made by the reviewer and we agree that the experimental results we present in our manuscript could have an impact in the field of integrated photonics in the long term. In particular, there has been a renaissance on research in Brillouin interactions in integrated photonics and visions for chip-scale radiofrequency signal processors based on strong on-chip optomechanical interactions have been proposed, such as in the following reference: [Eggleton, B. J., Poulton, C. G., Rakich, P. T., Steel, M. J. & Bahl, G. Brillouin integrated photonics. *Nat Photonics* **13**, 664-677, (2019)]. These interactions rely on the generation and detection of acoustic waves. Therefore, the acoustoelectric amplifier and circulator that we have developed could potentially be integrated in these systems to provide larger Brillouin gain and nonreciprocity for optical isolation. In particular, the gain in Brillouin processes is inversely proportional to the phonon loss rates; so the ability to directly reduce these loss rates would allow for significantly larger gain in Brillouin amplification and isolation. To address this point we have added the following to the Discussion section of the main text:

“Devices of this kind could also be utilized in future chip-scale systems for radiofrequency signal processing that use Brillouin interactions in integrated photonics. Given that the gain in Brillouin amplifiers and lasers is inversely proportional to phonon loss rates, the ability to actively control these (and reduce them to zero) could lead to unprecedented performance and novel functionality in these systems.⁵⁷”

Sincerely,

Matt Eichenfield, Ph.D.
Distinguished Member of Technical Staff
Sandia National Laboratories

Reviewers' Comments:

Reviewer #2:

Remarks to the Author:

This reviewer is satisfied with the authors' response and improvements to the manuscript and supplementary documents.

Reviewer #4:

Remarks to the Author:

The authors have addressed my minor comment. This is exciting work that should be published.

Reviewer 2:

1. “This reviewer is satisfied with the authors' response and improvements to the manuscript and supplementary documents.”

Answer to the reviewer:

We thank the reviewer for their review of our manuscript.

Reviewer 4:

1. “The authors have addressed my minor comment. This is exciting work that should be published.”

Answer to the reviewer:

We thank the reviewer for their review of our manuscript.

Thank you again.

Sincerely,

Matt Eichenfield, Ph.D.
Distinguished Member of Technical Staff
Sandia National Laboratories